# Research Status and Prospect of Laser Scribing Process and Equipment for Chemical Milling Parts in Aviation and Aerospace

**DOI:** 10.3390/mi13020323

**Published:** 2022-02-18

**Authors:** Jian Wang, Qiang Liu, Pengpeng Sun, Chenxin Zang, Liuquan Wang, Zhiwei Ning, Ming Li, Hui Wang

**Affiliations:** 1School of Mechanical Engineering and Automation, Beihang University, Beijing 100083, China; speng@buaa.edu.cn (P.S.); zcx2017@buaa.edu.cn (C.Z.); liuquanwang@buaa.edu.cn (L.W.); yijiu975272@buaa.edu.cn (Z.N.); 2Research and Application Center of Advanced CNC Machining Technology and Innovation, Beijing 100191, China; 3Jiangxi Research Institute, Beihang University, Nanchang 330096, China; 4Xi’an Institute of Optics and Precision Mechanics of CAS, Xi’an 710119, China; liming@opt.ac.cn; 5AECC Shenyang Liming Aero-Engine Co., Ltd., Shenyang 110043, China; wang13940542280@163.com

**Keywords:** laser scribing, chemical milling, ablation mechanism and technology, second scribing, selective laser removal process

## Abstract

Laser scribing in chemical milling is an important process which can effectively improve the precision and efficiency of chemical milling, and is of great significance to improve the thrust–weight ratio and manufacturing efficiency of aviation and aerospace parts. According to the scribing requirements in chemical milling for aviation and aerospace parts, the process and mechanism of laser scribing were studied and the influence of different process parameters for the quality of laser scribing was analyzed. Based on the review of related research literature, the laser scribing process, the ablation mechanism and technology of different materials and the selective laser removal process for “laser–coating–substrate” are summarized and discussed. Based on the requirements of high-precision laser scribing on complex surfaces, the current situation of laser scribing equipment is summarized. Finally, the practical challenges and key technical problems for the laser scribing process are summarized, and the application and development of laser scribing in aerospace manufacturing are prospected.

## 1. Introduction

With the continuous development of aerospace, the design and manufacture of lightweight components have become the key issues at this stage under the premise of ensuring high performance and reliability in aerospace [1]. In order to solve this problem effectively, a large number of thin-walled parts have been designed in aerospace vehicles, including aircraft engines and aeroengine casing, aircraft large skin parts, rocket tank wall plates, aluminum alloy skin serrated shallow step structures, aircraft wings, etc. The typical characteristics of these parts are a thin wall, large size, high design accuracy and complex features and structures [2,3]. The manufacture of these parts is extremely difficult and is an internationally recognized as a complex manufacturing problem, which poses great challenges to existing manufacturing technologies and equipment [4,5].

At present, the thin-walled aerospace parts are chemically milled after traditional metal cutting, in which the excess metal on the surface of the parts is removed by chemical milling to form features such as reinforcing ribs and structural bosses [6,7,8]. Chemical milling is a non-traditional machining process where a chemical solution is used to selectively remove material from a workpiece using a strong chemical reagent known as etchant [9,10]. Before chemical milling, the surface of parts should be coated with maskant. Then, some maskant is trimmed away and the uncovered area is subjected to the etching. The maskant could be cut by mechanical cutting or by laser ablation, which is called the scribing process in chemical milling [11,12]. For a long time, the scribing process in chemical milling has been done manually with a thin scalpel. There are many defects and difficulties in manual scribing for large complex surface thin-walled parts, such as poor precision, high cost, long cycle, difficulty with second engraving, etc. The scribing processes in chemical milling has become one of the key restrictions for manufacturing and chemical milling of aviation equipment parts [13].

The scribing process in chemical milling is essentially the selective removal of maskant materials on the surface of aviation and aerospace structures [14]. The existing technologies include: physical scribing [15,16], laser scribing [11], micro-milling scribing [17], micro-hot pressing [18], ultrasonic machining [19], plasma scribing [20,21], electron beam scribing [22], high-energy water jet [7], etc. Aircraft manufacturers in the United States and the European Union have been using lasers to engrave patterns on chemical milling parts for a long time, to solve the process problem of laser scribing on the surface of thin-walled parts and to accomplish the laser scribing of 3D skin parts with large curvature. The research of laser scribing in China is still in its infancy, and is mainly focused on the selection of laser scribing parameters for plane parts and the first scribing of large curvature surfaces by using different laser scribing machines. There are few studies on the laser scribing process and equipment for thin-walled parts with complex curved surfaces and various shapes of 3D structures, and there are still many unsolved technical problems.

The research on the mechanism, process and equipment of laser scribing for the 3D complex surface aerospace parts has great significance for manufacturing chemical milling parts. In this paper, the complex process requirements and laser scribing mechanism for laser scribing of aeroengine casings are analyzed in Section 2. The research status of the laser scribing process, laser ablation process and the selective laser surface removal process is summarized in Section 3. The research situation of laser scribing machines is studied in Section 4. The future development and application of laser scribing are prospected in the last section.

## 2. Process and Mechanism of Laser Scribing

### 2.1. Definition of Laser Scribing Process

Laser scribing in chemical milling was developed from the traditional manual scribing, which is an important process in the chemical milling of aerospace parts. In essence, it is a technological process of laser ablation for a chemical milling protective coating on the surface of aluminum or titanium, which is to engrave patterns on the surface of parts and ensure that the metal substrate material is not damaged. The laser scribing combines a laser with multiaxis CNC machining to make the scribing efficient and precise, and is programmed based on geometric pattern information and process parameters of chemical milling [23]. Compared with manual scribing, it can greatly shorten the production time, improve the dimensional accuracy and ensure the consistency and reliability of products [9,24]. The laser scribing is shown in Figure 1.

The main advantages of laser scribing include [25,26,27]:

(1) High accuracy. Laser scribing can improve dimensional accuracy, and the minimum width of a slit can reach 0.01 mm. The heat-affected zone of laser scribing is small, the workpiece deformation is small and there is no machining stress. The laser power in a certain range has no effect on the surface of metal parts, which ensures that the protective maskant can be cut off reliably without damaging the metal parts.

(2) High efficiency and flexibility. Compared with the traditional scribing, laser scribing can shorten the production time. It is not only suitable for small batch development of parts, but also suitable for mass production.

(3) Reduced cost and labor intensity. Since the scribing template is no longer used, the laser scribing can reduce the cost by saving a lot of materials used for manufacturing the template, and the labor intensity for the operator can be significantly reduced.

(4) Realizing complex surface scribing. It is necessary to accurately, efficiently and repeatedly engrave the various features on complex structural parts, which is also the best solution for second scribing.

### 2.2. Laser Scribing Process Requirements

According to the manufacturing of a typical aeroengine casing, the outer contour of the casing is covered with various geometrical mountings and a large number of reinforcing ribs which are designed as “I-shape” or “T-shape” structures to maximize weight reduction while ensuring the strength and rigidity of the casing [28]. The manufacturing features of aeroengine casing parts are shown in Figure 2.

The requirements of the scribing process are as follows: (1) the maskant shall be cut off without damaging the substrate; (2) the trajectory of the scribing pattern shall be continuous and smooth; (3) on the outer wall of the cylinder, the horizontal direction and normal direction of the cylinder wall can be engraved according to the figure; (4) the second scribing needs to be aligned with the edge of the first engraved shape; (5) the accuracy of the second scribing should be ±0.05 mm.

The chemical milling process of engine casing includes: chemical degreasing, sand blowing, coating maskant, scribing, uncovering maskant and chemical milling [29,30]. As an important process in chemical milling, the scribing process has an important effect on the quality of part milling. The laser scribing process of the engine casing is as follows [31]: (1) the casing blank is coated with maskant after chemical degreasing and sand blowing; (2) first scribing according to the engraved pattern; (3) first chemical milling; (4) the casing is coated with maskant again; (5) second scribing according to the engraved pattern; (6) second chemical milling; (7) obtaining the final part. The chemical milling process of aeroengine casing parts is shown in Figure 3.

In the scribing process of aeroengine casing, there are obvious differences in the actual patterns between the first and second scribing. The laser beam is generally perpendicular to the surface of the cylinder in the first scribing, and the features of the casing surface are engraved according to the pattern shown in Figure 2b. The second scribing is on the side wall of the stiffener which forms the approximate arc surface after the first chemical milling. In the second scribing, the laser beam must be inclined to scribe the side wall of the stiffener section, as shown in Figure 4.

The stiffener interface of parts is shown in Figure 5 after laser scribing and chemical milling twice. The cross-section of the stiffeners when measured after first scribing and chemical milling is shown on the right side of Figure 5. The cross-section of the stiffeners when measured after second scribing and chemical milling is shown on the left side of Figure 5. 

### 2.3. Mechanism and Process Parameters of Laser Scribing

#### 2.3.1. Laser Scribing Mechanism

The laser scribing is a laser application based on laser ablation, which irradiates a high-energy laser beam on the maskant of parts to remove the maskant on a metal surface without damaging the substrate. Due to the interaction between the laser and the material, there are many physical and chemical reactions on the surface of the material in laser scribing at the same time, including ablation, decomposition, ionization, degradation, melting, combustion, gasification, vibration, splash, expansion, shrinkage, explosion, peeling, and others [32]. The main mechanism of laser scribing is the ablation effect, which is the thermal effect generated by the high-energy laser to eliminate the combination of the maskant and the substrate by evaporation, plume, ionization and explosion [33]. The mechanism of laser scribing is shown in Figure 6.

The chemical milling maskant is a protective coating which can resist corrosive solution [34]. There are four common maskants, AC850, YT-5100, HH968-2 and KBL302D. The main components of the maskant for chemical milling are tetrachloroethylene, toluene, xylene, ethyl benzene, talc and other substances. Therefore, there are both organic and inorganic substances in the maskant. At the same time, the maskant evolves into various types of compounds under the action of spraying, oxygen and ultraviolet rays in the air [35].

Generally, the ablation threshold of the metal is between 1 J/cm^2^ and 10 J/cm^2^. The ablation threshold of the inorganic insulator is between 0.5 J/cm^2^ and 2 J/cm^2^. Additionally, the ablation threshold of the organic substance is between 0.1 J/cm^2^ and 1 J/cm^2^. The melting point of pure titanium is 1668 °C, the modification temperature of talc is 800 °C and the melting point of organic matter is generally 100–800 °C. 

The laser scribing quality under argon protection/no argon protection is compared for AC850 protective adhesive. The results show that argon protection in laser scribing has no significant effect on laser scribing quality [36], as shown in Table 1. Therefore, the maskant with a high laser energy absorption rate is rapidly processed to the melting point for gasification in the laser scribing process, while the titanium alloy substrate will not melt.

Through the analysis, the main mechanism of laser scribing is laser selective gasification, which is based on the difference in the absorption coefficient of the beam energy between the maskant and the substrate.
(1)$$F_{0}=(1-R)I_{0}=AI_{0}$$
where: $F_{0}$ is absorbed fluence (energy per area); R is reflectivity; $I_{0}$ is laser fluence; A is absorbance.
$$A_{1}I_{0}<T_{1}$$
(2)$$A_{2}I_{0}>T_{2}$$
where: $A_{1}$, $A_{2}$ are absorbance of the substrate and chemical milling maskant, respectively; $T_{1}$, $T_{2}$ are the melting temperature of the substrate and the melting temperature of chemical milling maskant, respectively.

When the properties of metal substrate and maskant meet those of Formula (2), the energy absorbed by the metal substrate has not reached the melting temperature, and the energy absorbed by the maskant has reached the gasification temperature to remove the maskant on the surface. When the laser energy is greater than the ablation threshold of metal materials, the damage to the metal substrate cannot be avoided, and it is necessary to control the laser process parameters to minimize the damage.

#### 2.3.2. Laser Scribing Process Parameters

Due to the diversity of the maskants, the interaction of the laser ablation process between laser and maskant/substrate is complicated. Therefore, a suitable scribing model must be selected to achieve a good etching effect. The laser wavelength, the energy density, the pulse frequency, the pulse time and the incident angle of the laser must be considered comprehensively.

Laser wavelength

Laser scribing is mainly based on the interaction between the laser and maskant/metal substrate, which mainly depends on the material properties and laser wavelength [37]. According to related research, CO_2_ lasers are typically used for maskant scribing applications. Additionally, a CO_2_ laser is the most suitable laser to remove the organic material on the metal surface [38]. When the laser wavelength is 10.6 μm, the absorption coefficient of organic material is much larger than most metals [39,40]. When a CO_2_ laser is used to remove the organic matter on the metal surface, the temperature of the organic matter rapidly rises to the evaporation point, while the metal substrate not reach the melting temperature. Therefore, it is possible to remove the contaminated layer without damaging the metal substrate. In laser scribing, the mechanism of laser scribing also depends on the optical characteristics, thickness, laser energy density and thermal conduction effect of organic substances [41], and different types of lasers can also be used for scribing [42].

2.Laser power

In order to effectively improve the scribing quality, an appropriate laser power should be selected to match the ablation threshold of the material. 

The relationship between the ablation diameter and absorbed fluence is as follows [43]:(3)$$D^{2}=2\omega_{0}^{2}\ln\left(\frac{I_{0}}{I_{th}}\right)$$
where
$\omega_{0}$ is laser spot diameter;$I_{0}$ is laser fluence;$I_{th}$ is threshold fluence.

In particular, the maskant is mainly composed of colorless organic compounds and small amounts of dyes and additives. It is a kind of semitransparent material, which is suitable for the Lambert–Beer law. When the laser’s absorbed fluence exceeds the threshold fluence, the depth of the material removed by each pulse increases with the logarithm of the energy density, which is in accordance with the Lambert–Beer law.
(4)$$I\left(z\right)=\left(1-R\right)I_{0}e^{-\alpha z}$$
where
$I\left(z\right)$ is laser fluence in the target as a function of *z*;α is effective absorption coefficient;z is distance from material surface.

According to the Lambert–Beer law, the logarithmic relationship between the ablation rate (ablation depth) and the laser fluence can be obtained as follows [44]:(5)$$z\left(I_{0}\right)=\alpha^{-1}\ln\left(\frac{I_{0}}{I_{th}}\right)$$
where $z\left(I_{0}\right)$ is the ablation rate (ablation depth) as a function of $I_{0}$.

Laser fluence is an important factor affecting laser scribing quality. The morphology is inversely proportional to the laser fluence. When the laser fluence is low, there is a better morphology. When the laser fluence is increased, the morphology becomes worse. The occurrence of explosions can be avoided and the processing morphology can be optimized by selecting a lower laser fluence than the explosion threshold of the material phase.

It should be noted that the basic Lambert–Beer law applied in this paper is the result of analysis and simplification, and a more realistic form should be considered. In terms of a theoretical model, the absorption of laser energy on the material surface is affected by many factors, including laser transverse mode, absorption coefficient, reflection coefficient, surface roughness, temperature, multiphoton absorption, etc. The model parameters are simplified as follows: (1) the circular spot with Gaussian distribution is adopted. It is considered that the absorption coefficient of the laser in the horizontal direction is the same, and the laser reverse mode is constant. (2) The surface roughness of the maskant sprayed by the same process is basically the same, so the surface reflectivity is basically unchanged. Meanwhile, the second distribution of energy caused by thermal conductivity makes the whole volume of laser energy more uniform, and the non-uniformity of surface energy caused by roughness will be greatly weakened. Therefore, the model simplifies and ignores the surface effect of roughness on energy. (3) With the increase in temperature, the change in bond length displacement of the polymer is very small, and the change in formant in the process of CO_2_ laser energy absorption by the polymer is also very small, which can be basically ignored. Meanwhile, the surrounding heat diffusion is affected by the previous pulse energy, which will make the processed area increase in temperature by a certain amount. Under the current pulse energy, the temperature rise gradient from material heating to melting and vaporization changes little, and has little effect on the absorption coefficient. Therefore, the effect of temperature on the absorption of the material surface is also simplified and ignored. (4) The multiple multiphoton absorption coefficient is related to the instantaneous power density. The CO_2_ laser peak power is very small, which is not enough to produce multiphoton absorption. Therefore, laser scribing mainly considers the single photon absorption of materials. Based on the above analysis, the simplified theoretical model conforms to the basic Lambert–Beer law.

3.The incidence angle of the laser beam

The incidence angle of the laser beam has a great influence on laser scribing. With the change in the inclination angle of the laser, the laser scribing depth of the maskant becomes deeper (h2 > h1), and the radiation area of the laser becomes wider (S2 > S1), which is shown in Figure 7. At the same power density, the laser energy density of the scribing surface can be decreased and the depth of the scribing surface can be decreased with the increase in laser incidence angle. Therefore, the parameters of first laser scribing cannot be directly used for the second scribing. The systematic and mature theory about the influence of the angle of inclination on the laser scribing needs further research and verification.

4.Laser scribing speed

The scribing speed is also an important factor affecting the laser scribing quality. The influence of scribing speed mainly includes overlapping rate and the energy density. The scribing speed is inversely proportional to the laser overlapping rate, which will directly affect the processing quality. If the overlap rate is too low, the processed surface features will be discontinuous, and there will be wavy bulges, which are shown in Figure 8. If the overlap rate is too high, not only will the machining efficiency be reduced, but the laser energy injected at the same location will also be excessive, which will also produce poor machining morphology. The overlap rate is defined as follows:(6)$$O_{d}=1-\frac{v}{df}$$
where
v is laser scribing speed;d is spot diameter;f is repetition rate.
Figure 8Overlap rate and residual height of ablation width.
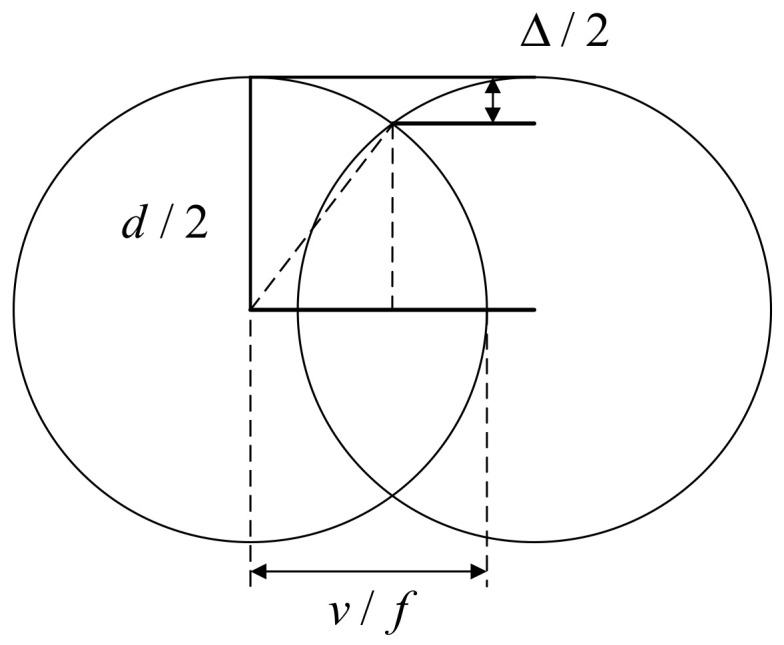


(7)1−(1−Δ)2=vdf
where
Δ is residual height of ablation width.

The influence of the scribing speed on the laser energy density is mainly reflected in: (1) when the scribing speed is fast, the accumulated laser energy per area is small; (2) when the scribing speed is slow, the accumulated laser energy per area is large.
(8)$$I_{0}=\frac{P}{v\times w}$$
where
$I_{0}$ is laser fluence;P is laser power;v is laser scribing speed;w is width of scribed line.

When the width of scribed line is equal to the spot diameter, w=d.

### 2.4. Key Research Points of Laser Scribing Process

In summary, the following technical problems need to be solved in the process of laser scribing and manufacturing [45,46]:
(1)Laser scribing process for chemical milling parts. The scribing model including laser parameters and motion parameters should be established based on scribing quality of different laser process parameters. Additionally, the process parameters of first and second scribing should be studied for the laser scribing of engine casing.(2)Ablation mechanism of different materials in laser scribing. Based on the maskant and substrate materials of laser scribing, the technology and model of laser ablation for organic polymer and metallic materials should be studied. By controlling the laser power, frequency, speed and other parameters, the shape of the scribing line on the part’s surface can be precisely controlled.(3)Laser selective removal process for “laser–maskant–substrate”. The process and evolution law of laser selective removal, the removal process for different film/substrate materials and the application of laser selective removal should be studied based on material physical–chemical properties and laser energy transmission characteristics.(4)Laser scribing equipment. According to the requirements of large size, complex feature surface and multiple scribing, a micro-kerf at the micron level and macro-pattern at the meter level should be realized in laser scribing. Laser scribing also needs to meet the requirements of accurate positioning and graphic etching on complex 3D surfaces, high processing and positioning accuracy and completing the etching of surface graphics in one run. Therefore, a five-axis CNC laser system with a large processing range and high processing accuracy must be used for laser scribing, which is to ensure that the laser head has a reasonable position and attitude in the laser manufacturing of 3D complex profiles.

## 3. Research Status of Laser Scribing Processes

In order to study the process and mechanism of laser scribing in chemical milling more comprehensively, laser scribing for chemical milling parts, the ablation mechanism of different materials in laser scribing, laser selective removal process for “laser–maskant–substrate” are reviewed and discussed, and are the theoretical basis for laser scribing processes to achieve the best quality and highest efficiency.

### 3.1. Laser Scribing Process for Chemical Milling Parts

As early as the 1970s, laser scribing was proposed and applied to semiconductor manufacturing [47]. With the continuous maturity of laser technology and equipment, laser scribing began to be applied to the chemical milling process in the 1980s [12]. Considering the particularity of the aerospace field, application manufacturers and scholars at home and abroad often adopt blockade and confidentiality, and there are few studies on the laser scribing process of chemical milling parts in aerospace.

In 1987, Gnanamuthu from Boeing North American Inc., Dallas, TX, USA [42] disclosed a method for scribing chemical milling maskant applied to a metal substrate by impinging a laser beam on the maskant and controlling the beam to penetrate through the maskant substantially without damaging the underlying metal. In the process, the Nd:YAG (wavelength = 1.06 μm), the Nd:Glass (wavelength = 1.06 μm) and CO_2_ (wavelength = 10.6 μm) laser beams controlled by NC or CNC were used to scribe the maskant with a thickness of about 10 mils (0.254 mm). In 1992, Slysh [48] proposed a laser-assisted masking process, which is a method of accurately applying maskant to a workpiece using one or more scanned laser beams and a maskant delivery device. The method includes the interaction of the laser beams with the maskant material while the material is in transit to, and after it has reached, the surface of the workpiece. In 2003, Griffin [49] studied the laser scribing of chemical milling maskant, and pointed out that laser power requirements for maskant scribing are determined by the number of optics in the beam path, and the desired feed rates for laser scribing. The light energy emitted from lasers in the 200–500 W range will reflect from aluminum substrate, thus there is no damage to the substrate. Lower laser power will cut the maskant, however, with a minimum of 200 W at the cutting surface, the laser scribing process can easily accommodate variations in maskant film thickness and variations in workpiece flatness. In 2010, Leone et al. [11] studied the application of laser ablation in chemical milling in the field of aeronautics and astronautics, and analyzed the influence of process parameters, cutting speed and beam power on the interaction phenomenon and defect formation in the polymer (maskant) laser cutting process under the condition of a CO_2_ laser source. In 2012, Cao et al. [15] analyzed the chemical milling structure of a titanium alloy welded case, and selected the optimal cutting parameters through multiple parameter laser cutting experiments to realize the flexible and accurate pattern scribing method of chemical milling graphics. Laser scribing process parameters and scribing quality are shown in Table 2. In 2020, Pei et al. [36] studied the relationship between laser power, scribing speed, laser frequency and scribing depth in process parameters, and pointed out that there is an obvious linear positive correlation between laser scribing power, scribing speed and scribing depth. The relationship between laser power, scribing speed and scribing depth is shown in Figure 9.

At present, there are some studies on the striping process, which is mainly focused on the testing and selection of the process parameters of laser scribing by using a CO_2_ laser. In order to meet the requirements of the laser scribing mechanism and process for case parts, further research needs to be carried out, including: research on the ablation threshold and absorption rate of different maskants for different lasers, research on the second scribing process, research on the damage of parts after laser scribing, ablation effects of machined parts, the laser scribing mechanism and evaluation of laser scribing quality.

### 3.2. Laser Ablation for Mechanism and Model of Chemical Milling Maskant

In order to further study the process and mechanism of laser scribing, the relevant mechanism and model of the laser ablation process for organic polymer and metallic materials are summarized based on the maskant and substrate materials of aerospace chemical milling parts [50,51].

The relevant mechanism and model of the laser ablation process for organic polymer have been established by scholars [52,53,54]. Arnold et al. [55] described the ablation of organic polymers on the basis of photothermal bond breaking within the bulk material. By assuming a first order chemical reaction chemical reaction meets the Arrhenius law, the ablation starts when the density of broken bonds at the surface reaches a certain critical value, which explains the ablation behavior near the threshold value. Nakamura et al. [56] investigated the ablation characteristics of tetrafluoroethylene-hexafluoropropylene copolymer (FEP) film with femtosecond and picosecond lasers. Meanwhile, the ablation characteristics of femtosecond and picosecond titanium sapphire laser pulses at 798 nm were studied, and the effects of different pulse widths on the ablation rate were analyzed. Dyer [57] studied the mechanism and application of laser ablation, and analyzed the photon absorption in the smoke area of laser etching polymer materials, which reduced the effective ablation density of the material surface, and affected the improvement of the surface etching speed precision. Escobar [58] investigated the laser ablation of polymers for micro-channel fabrication and selective laser melting of metal powders, and presented the predictive modeling, simulation and optimization of laser processing techniques. Desai and Shaikh [59] investigated micro-milling performance of thermoplastics with different parameters, including laser beam absorptivity, latent heat of vaporization, laser power and cutting speed. Fifty different combinations of laser power and cutting speed with four categories of thermoplastics, namely poly-methyl-methacrylate, poly-propylene, acrylonitrile butadiene styrene and nylon 6, were studied by using a 25 W CO_2_ laser scribing machine. Antończak et al. [60] established a theoretical model of pulsed CO_2_ laser grooving for modified nylon, and presented the relation model of laser beam intensity, pulse repetition rate and material scanning speed in the basic parameters of laser grooving technology. Arnold et al. [55] performed ablation experiments with ultrashort laser pulses (pulse duration 150 fs, wavelength 800 nm) on polymers (PC, PMMA) relevant for biomedical technology, and proved that there is a significant relationship between the ablation threshold and the number of pulses by introducing the accumulation mode. Sauerbrey and Pettit [61] studied the theory about the etching of organic materials by ultraviolet laser pulses, and pointed out that there is an equivalent condition for the laser ablation threshold. When the number density of photons absorbed is equal to the density of chromophores, it indicates that the threshold condition has been reached. Whiting et al. [62] found the evolution of the ablating surface, and formulated a mathematical model of ablation of tissue by an incident laser beam in terms of a series of layers in which laser power is absorbed and water and tissue vaporized. Zhanet et al. [63] proposed a facile and rapid method to obtain outstanding superhydrophobic surfaces by CO_2_ laser micro-processing of a PTFE plate.

There are many studies on laser scribing and ablation technology of metal materials. Pramanik et al. [64] investigated laser marking on aluminum 6061 alloy by a fiber laser. Experimental analysis on the basis of response surface methodology (RSM) has been carried out for determination of a mathematical model. Additionally, an optimization analysis on marking quality, mark width and mark depth has been carried out. Lu et al. [65] studied the application of laser marking on a magnesium alloy surface, and analyzed the influence of laser type, laser scanning speed, laser power, light spot diameter and other parameters combined with surface morphology characteristics. Wang and Zeng [66] investigated the mechanism of laser scribing, and established a mathematical model of the relationship between laser processing parameters and laser scribing depth. Zhao et al. [67] put forward the collaborative control technology of feed speed and laser power for complex curved surface patterns, considering the dynamic characteristics of machine tools. The ablation depth is used to characterize the removal rate, and a nanosecond multipulse laser ablation depth prediction model based on the thermal energy balance principle was established. By using the characterization of ablation depth to the removal amount, a nanosecond-level multipulse laser ablation depth prediction model based on the principle of thermal energy balance was established. Yue et al. [68] reported the feasibility and characteristics of a short-pulsed laser to remove an oxygen-enriched alpha case layer from a titanium alloy (Ti6Al4V) substrate. The material removal rate, ablation rate and ablation threshold of the alpha case titanium were experimentally determined, and compared with those for the removal of bulk.

In summary, due to the complex laser action mechanisms of organic polymer compared to metallic materials, it is common to seek optimal process parameters by using experimental or theoretical modeling methods. There are many studies on laser ablation mechanisms and models for organic polymers and metals, which can provide an effective reference for the application of laser scribing. The existing laser ablation application is different from laser scribing, which is mainly focused on micro-structures of parts with planes and large curvatures.

### 3.3. Laser Selective Removal Process for “Laser–Maskant–Substrate”

Laser scribing is also a laser selective removal process, which is similar to laser cleaning and laser surface film manufacturing [69,70]. Laser cleaning is removing the contaminants on the surface of the parts [71]. Laser manufacturing for surface films is using a laser to manufacture on surface films of different substrates to achieve practical application requirements.

The application of laser cleaning was first proposed by the American scientist John Asmums in the early 1970s. In 1974, Fox [72] investigated surface cleaning on resin, glass and metal substrates. In particular, the removal of the coating layer on aircraft surfaces has been widely studied by researchers. Tam et al. of IBM has conducted in-depth research on the mechanism of laser cleaning, and proposed wet laser cleaning for the first time [73]. Kopf et al. [74] evaluated paint removal techniques and protective coatings for Air Force aircraft, with a focus on graphite–epoxy composite substrates. Chen et al. [75] studied the laser cleaning of marine paint on low-carbon steel, which is successfully used to remove the oil and gas layer on the surface and obtain the best process parameters. Kan [76] studied the cleaning of denim fabric by using a CO_2_ laser, and extended the application of laser cleaning to a new field. Schweizer and Werner [77] studied paint removal from aircraft surfaces by using a 2 kW TEA CO_2_ laser, and completely removed the paint with no damage to the metal substrate. Pantelakis and Kermanidis [78] studied paint stripping processes including laser radiation with excimer, CO_2_, TEA CO_2_ and YAG laser sources as well as plasma etching. These processes have been applied for the removal of polyurethane coating which is a typical aeronautical paint system. Daurelio et al. [79] found that the same object can be cleaned by different lasers and different objects can be cleaned by the same laser. The decisive factor depends on the different laser parameters required in the cleaning process. The United States Air Force and environmental safety technology certification program jointly conducted a study on the paint removal performance of hand-held laser paint removal equipment on metal and non-metal surfaces [80]. Hong et al. [81] studied the surface removal of unidirectional carbon fiber/epoxy resin composite coatings with different wavelengths of a power-modulated continuous laser. The companies Cleanlaser and Slcr in Germany have developed equipment to remove paint on the surface of composite materials and to remove damaged materials during repair [82]. Lee et al. [83] carried out laser removal of small particles from a metal surface by changing the incident angle of the laser beam. It has been found that a dramatic improvement of cleaning efficiency in terms of area and energy is observed when using the laser is at a glancing angle of incidence as compared to a perpendicular angle. Ahn et al. [84] analyzed the laser cleaning process for removing lubricating oil from metal surfaces. In an experiment, mineral oil was removed from carbon steel, stainless steel or copper surfaces using near-infrared (Nd:YAG) laser pulses that are weakly absorbed by the oil. The removal mechanism has been found to depend critically on the optical properties of the oil. Tang et al. [85] experimentally established the effects of sulfide layers and fluence values on the mechanism of laser cleaning. Ye et al. [86] studied 1064 nm laser-induced plasma shockwave cleaning, which is utilized to remove SiO_2_ particle contaminants on K9 glass surfaces. The effects of parameters (particle position, laser gap distance and laser energy) on the cleaning efficiency have been studied in the case of single pulse laser cleaning. Yokozeki et al. [87] applied pulsed laser surface treatment to carbon fiber-reinforced plastics (CFRPs) to enhance the work efficiency as well as to establish the stable cleaning process of surface pre-treatment during the bonding process of CFRP structures. Davarpanah et al. [88] and Li et al. [89] systematically reviewed the progress of methods including operation parameters, influence on substrates and stripping mechanism.

Laser manufacturing of surface films has also been widely applied and studied. Ryu et al. [90] successfully applied an ArF excimer laser to selectively clean ZnSe film on a GaAS surface. Ko et al. [91] explored the ablation of gold nanoparticle films on a polymer using a nanosecond pulsed laser, and demonstrated high resolution and clean feature fabrication with little energy and selective multilayer processing. Marimuthu et al. [92] reported removal of Ti-N coating on a WC tool by a pulsed laser. The laser energy density was determined to be 2 J/cm^2^, which satisfies the condition that the cleaning threshold is larger than the cleaning threshold (1.62 J/cm^2^) of Ti-N coating and smaller than the threshold (2.36 J/cm^2^) of the substrate WC. Takeshita et al. [93] demonstrated reflectivity adjustment of a film-coated laser facet using ablation etching. Li et al. [94] studied the ablation effect of nanosecond pulse laser parameters (pulse energy, pulse repetition rate and pulse number) on Al-Si coating, and obtained the calculated ablation threshold of fully removed Al-Si coating. Li and Xiong [95] studied the high-precision and non-destructive etching for composite/metal film materials, and observed the oxidation and ablation process of the film in the air by XRD and SEM. Farid et al. [96] investigated fast selective patterning with high precision on a 175 nm ITO thin film with IR pulse lasers. Gakovic et al. [97] investigated the interaction of ultrashort laser pulses with a titanium/aluminum (Ti/Al) nano-layered thin film. The ablation mechanism and the ablation threshold of Ti/Al nano-layered thin film under the condition of low and high laser energy were analyzed. Romoli et al. [98] investigated through-the-thickness laser ablation characteristics of ceramic coating. The effects of energy density, hatch distance and coating color on the ablation completion index were analyzed. During the through-the-thickness laser ablation of the ceramic coating, the input energy density of black ceramic coating should be in the range of 0.049–0.251 J/mm^2^, and the input energy density of the other coatings should be 0.112–0.251 J/mm^2^. Koziol et al. [99] presented an alternative method of manufacturing SRR structures through the selective removal of a thin layer of silver–palladium deposited on the surface of AlO ceramic by the laser ablation process using a nanosecond Nd:YAG laser (1064 nm).

Meanwhile, in the laser surface selective removal of 3D part surfaces, the machining of micro-structures on complex surfaces has been realized [100]. Meijer et al. [101] studied the optimal process parameters of laser ablation for polymer materials, and made polymer and tantalum vascular scaffolds with an ultrashort pulse laser. Kathuria [102] made miniature vascular stents by laser processing on the basis of a thin metal tube. Leisten et al. [103] proposed laser-assisted mask etching technology to solve the problem of miniaturization of communication equipment such as GSM, Bluetooth, wireless LAN, GPS, etc. Soulard et al. [104] proposed H-dimensional holographic lithography by improving planar lithography, which was used to manufacture a cone-shaped spiral antenna. Ye et al. [105] studied a laser etching method and laser film forming method, which were used to make conical micro-strip antennae based on polytetrafluoroethylene, polyimide and epoxy resin, respectively.

Based on the above analysis, the laser surface selective removal process for “laser–coating–substrate” has been developed for decades and has been widely used in engineering. The research mainly focuses on the influence of different laser process parameters (laser wavelength, laser power, scanning speed, frequency, duty cycle, number of scans, defocus amount, line spacing) on the protective film and metal substrate. The laser surface selective removal process has accomplished the cleaning of grease, paint, dust, embroidery and residual solvents, adhesives on the surface of steel, aluminum–magnesium alloy, titanium alloy, rubber plastic, silicon wafer, stone, cloth and other materials. It has been applied in the manufacture of some metal films, polymer films, composite material coatings and ceramic coatings, which provides a theoretical basis for the study of laser molding technology. For the application of 3D surface selective removal for complex surfaces, it has accomplished the ablation processing and manufacturing of micro-circular tubes, helical antennae and other parts. The existing 3D laser ablation application is different from laser scribing, which is mainly focused on micro-structures of parts with planes and large curvatures.

## 4. Laser Scribing Equipment

Laser scribing equipment is the final execution unit to realize laser scribing processing. In order to ensure the high efficiency and high precision of laser scribing on 3D arbitrarily complex curved surfaces, the design of laser scribing equipment needs to consider the requirements of the part size, complex surface features and multiple scribing process of the aeroengine casing [106,107]. Based on the current situation of laser scribing equipment, the research status of 3D laser scribing equipment is reviewed in this paper.

### 4.1. Review of Laser Scribing Machine

The scribing process was automated with the introduction of a laser scribing machine in the early 1980s at McDonnell Douglas in St. Louis, MO, USA. The machine was designed and constructed to scribe flat aluminum sheet stock only. More sophisticated multiaxis laser scribing machines were developed at the Douglas Aircraft Division of McDonnell Douglas in the mid-1980s [49]. Since the 1980s, the Spanish company M. Torres has been designing, manufacturing and installing five-axis gantry laser scribing machine systems with flexible fixtures, which were the first laser scribing machines system of Airbus. After 30 years of development, a series of products have been formed which can adapt to various shapes of chemical milling parts. An automatic laser maskant scriber with linear synchronous motors (LSMs) was built in 1993 for Boeing, Wichita, KS, USA [108], which is used to scribe the maskant of aircraft aluminum skin panels in chemical milling. LSMs rapidly and precisely position a huge gantry with the CO_2_ laser maskant scribing system. According to Boeing, this is the largest automatic laser scriber in the world (33 m long, 3 m wide and 5.1 m tall) and can hold two Boeing 747 wings simultaneously, scribing one while positioning the other, and scribe up to 2.54 m/s or complete up to four skin panels in 1 h [109,110]. Sarh et al. [111] introduced a five-DOF gantry robotic system and flexible pogo fixture using a carbon dioxide laser to scribe the mask, enabling mask removal in certain skin locations. Meanwhile, the laser scribing equipment developed by M. Torres and Prima has been used by Boeing, Airbus and other aircraft manufacturing enterprises to achieve the manufacturing of aeroengine cases, blades and aircraft skin surfaces. The existing laser scribing equipment in China is mainly the laser scribing machine developed by Prima and M. Torres. The relevant equipment has been introduced to China’s aircraft manufacturers and has been applied in practice. The Xi’an Institute of Optics and Precision Mechanics of CAS has developed laser scribing equipment which can meet the requirements of thin-walled part milling, and has been used for engineering applications in aeroengine manufacturing enterprises.

The comparative analysis of laser engraving machines in the above literature is shown in Table 3.

Generally, the laser scribing machine is similar to the five-axis laser machine, which is composed of three linear axes and two rotary axes. Laser scribing machines are now available from virtually any manufacturer of laser machine tools. The common structures of 3D laser equipment are vertical, horizontal and gantry types, and its configuration is similar to that of general machine tools. In addition, the multiaxis series-parallel manipulator structure is also more used in 3D laser equipment [112]. In order to meet the different application requirements of laser processing, different laser machines have been proposed. The German company DMG has developed a series of laser machine tools named LASERTEC, which are used for the laser texture etching of mold surfaces. In this laser machine, a two-dimensional scanning galvanometer is installed on the five-axis CNC machine tool instead of a tool system. The DWL400 laser direct writing system [113] was proposed by Fraunhofer Institute Laser Technik to make the surface of the substrate and the photoetching objective always vertical by synchronously cooperating with the rotation of the optical platform, so as to realize the real-time centering and focusing operation, and then complete the direct writing of the graphics. Gray et al. [114] studied the five-axis laser galvanometer scanning system to realize two-dimensional milling by two galvanometer shafts, two-dimensional milling by three mechanical shafts and two galvanometer shafts and three-dimensional milling by five mechanical shafts and two galvanometer shafts. Diaci et al. [115] reported a laser marking system with a “3 + 2” structure, and integrated a continuous laser scanning monitoring system to achieve the etching of small-scale curved metal surfaces. Chen [116] also reported a laser machine tool with 3D mechanical shafts and 2D galvanometer shafts, which realized laser etching on the surface of an eggshell. Jiang et al. [117] studied the reconfigurable structure of the integration of an ordinary high-speed mechanical gantry and laser processing head, and realized the layered surface texture system of a large free-form surface workpiece. Lianghui et al. [118] studied five-axis 3D laser equipment which is composed of a 3D laser galvanometer system and a two-axis numerical control rotary table. Wang et al. [119] studied “5 + 3” axis laser etching equipment composed of a five-axis CNC machine tool and three-dimensional scanning head, which provided high-precision, high-quality and high-efficiency machining of the surface functional structure of complex curved parts.

### 4.2. Several Typical Scribing Machines

#### 4.2.1. TORRESLASER Laser Scribing Machine

The TORRESLASER laser scribing system integrated with the TORRESTOOL fixture system became the first scribing equipment at Airbus’s German factory. Some of the TORRESLASER laser scribing equipment parameters are shown in Table 4.

#### 4.2.2. LASERDYNE-890 Fiber Laser System

The LASERDYNE-890 fiber laser system is a five-axis laser processing system for large part welding, drilling and cutting processes. After the appropriate modification of laser process parameters and program optimization, the high-precision and high-efficiency scribing of complex lines on the coating can be realized. Based on the characteristics of high reflection rate of metal and high absorption rate of non-metal using a CO_2_ laser, cutting the protective coating and not burning titanium alloy parts are realized. The system has been used in the laser cutting of difficult materials for aeroengines, the group hole machining of large thin-walled parts, the blade hole cutting of parts and the laser scribing of thin-walled parts. Among them, the basic parameters of the LASERDYNE-890 processing system for laser scribing are as follows: X/Y/Z-axis travel is 2400 mm × 1800 mm × 900 mm; the system is equipped with a 3000 W CO_2_ laser and five-axis linkage full flight optical path.

#### 4.2.3. Laser Scribing Machine of Xi’an Institute of Optics and Precision Mechanics of CAS

A six-axis with five-axis linkage laser scribing machine has been developed by the Xi’an Institute of Optics and Precision Mechanics of CAS, which has been applied in Shenyang Liming Aero Engine Co., Ltd. and has the actual processing functions of first and second scribing. The parameters of the laser scribing machine are shown in Table 5. The design diagram of the laser scribing machine is shown in Figure 10.

#### 4.2.4. Laser Scribing Machine Designed by the Research Team of the Authors

In addition, according to the requirements of the scribing process, the research team of the authors also designed a principle prototype of a six-axis five-linkage intelligent NC laser scribing machine, which is shown in Figure 11.

### 4.3. Comments

To summarize the application of laser scribing machines, the existing laser scribing machines can basically meet the requirements for first scribing, and can carry out the 3D machining of aircraft skin parts with large curvatures. The laser scribing machines developed by the Xi’an Institute of Optics and Precision Mechanics of CAS accomplished the test verification of the first and second scribing, and the actual engineering application needs further verification. It should be noted that laser scribing machines for functions such as multiparameter and multivariable coupling control of the second scribing process, second clamping positioning, part deformation detection and complex surface laser scribing trajectory generation have not been found in the relevant literature, which means that the related technology difficulties have not been completely resolved.

## 5. Summary and Prospects

In this paper, the laser scribing process and equipment for chemical milling parts in aerospace are introduced. Laser scribing as an efficient method, can quickly and effectively remove the maskant, greatly shorten the manufacturing time and guarantee the manufacturing quality in chemical milling. This paper studies the process and mechanism of laser scribing, and analyzes the influence of laser wavelength, laser power, scribing speed and other process parameters on the quality of scribing, which provides a theoretical basis for the research of special technology of laser scribing.

In terms of the current situation and problems of laser scribing technology, a lot of research has been carried out on the existing laser removal technique and laser surface scribing technique. The research on the laser scribing for chemical milling parts mainly focuses on the selection and experimental verification of the first scribing parameters. The theory and method of laser scribing modeling, the law of mesoscale evolution based on the interaction of “laser–coating–substrate” and the selection of motion parameters with laser parameters for 3D complex surfaces need to be further tested and studied.

The existing 3D laser equipment has been able to accomplish the laser processing for 3D complex profile parts, which is mainly focused on cutting, welding, marking and other fields. The laser scribing machine developed by Prima and M. Torres can basically accomplish the first scribing for small curvature surface parts, and there is no relevant literature proving that it can be used for the second scribing. The laser scribing equipment in China began to be studied and developed, and has accomplished first and second scribing, and has been used in engineering applications.

With the research of laser scribing technology and equipment, laser scribing technology will play an increasingly effective role in the manufacture of aerospace engine casing parts. The key technologies of the laser scribing process and equipment development are not limited to those mentioned above. The common key technologies in the manufacturing field, such as intelligent manufacturing technology, open module structure and ubiquitous sensing technology, can also be integrated in the laser scribing equipment, and finally form a high-speed, high-precision, green intelligent multiaxis NC laser manufacturing unit.

## Figures and Tables

**Figure 1 micromachines-13-00323-f001:**
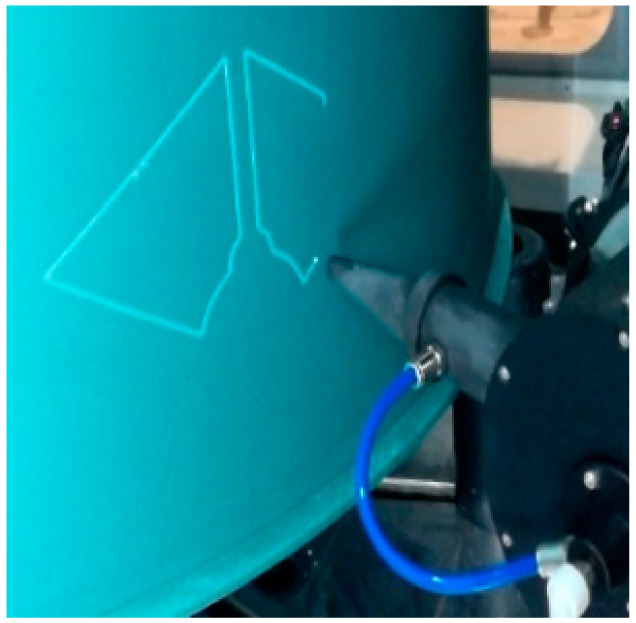
Laser scribing.

**Figure 2 micromachines-13-00323-f002:**
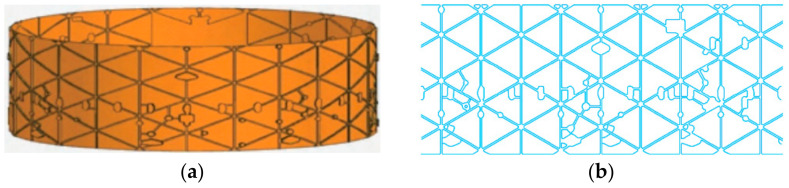
Manufacturing features of aeroengine casing parts. (**a**) Aeroengine casing parts; (**b**) manufacturing features of casing.

**Figure 3 micromachines-13-00323-f003:**
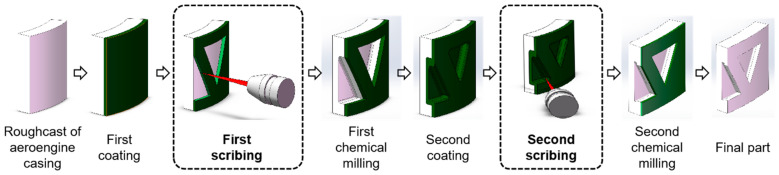
Manufacturing process of aeroengine casing parts.

**Figure 4 micromachines-13-00323-f004:**
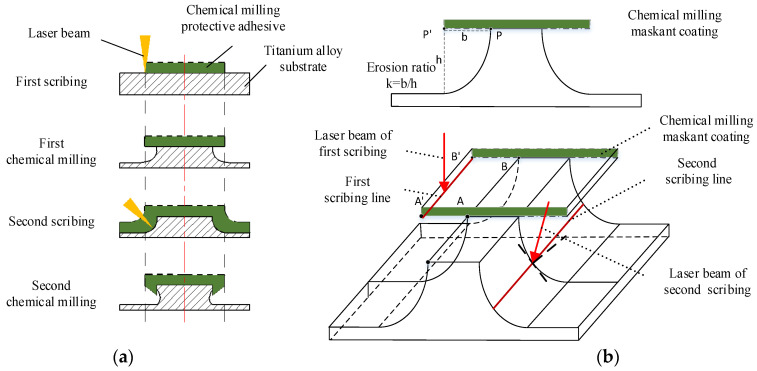
Section diagram of laser scribing. (**a**) Schematic diagram of stiffener for laser scribing; (**b**) schematic diagram of twice laser scribing.

**Figure 5 micromachines-13-00323-f005:**
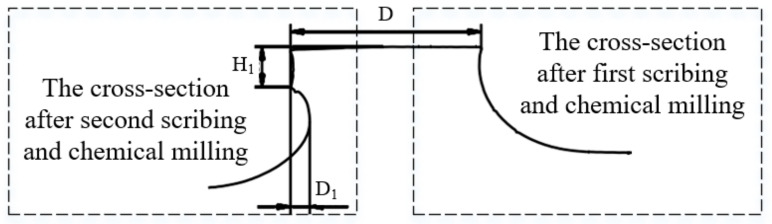
Cross-section characteristics of stiffening ribs of casing parts.

**Figure 6 micromachines-13-00323-f006:**
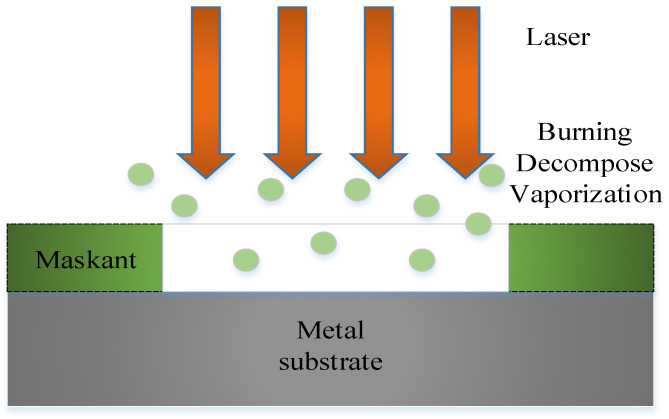
Mechanism of laser scribing.

**Figure 7 micromachines-13-00323-f007:**
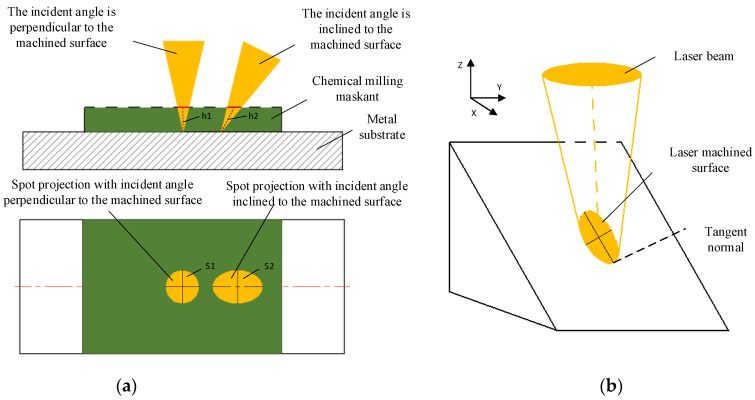
The influence of laser beam incidence angle. (**a**) The influence of laser beam incidence angle on depth and area; (**b**)Influence of laser beam incidence angle on depth and area from 3D view.

**Figure 9 micromachines-13-00323-f009:**
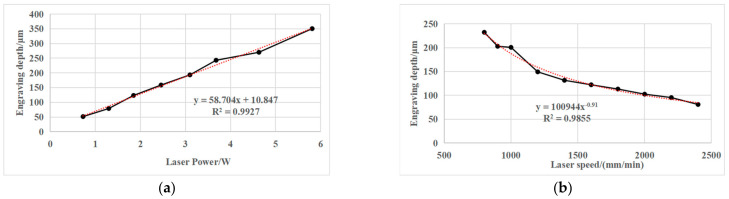
Relationship between laser power/speed and scribing depth. (**a**) Relationship between laser power and scribing depth; (**b**) Relationship between laser speed and scribing depth [36].

**Figure 10 micromachines-13-00323-f010:**
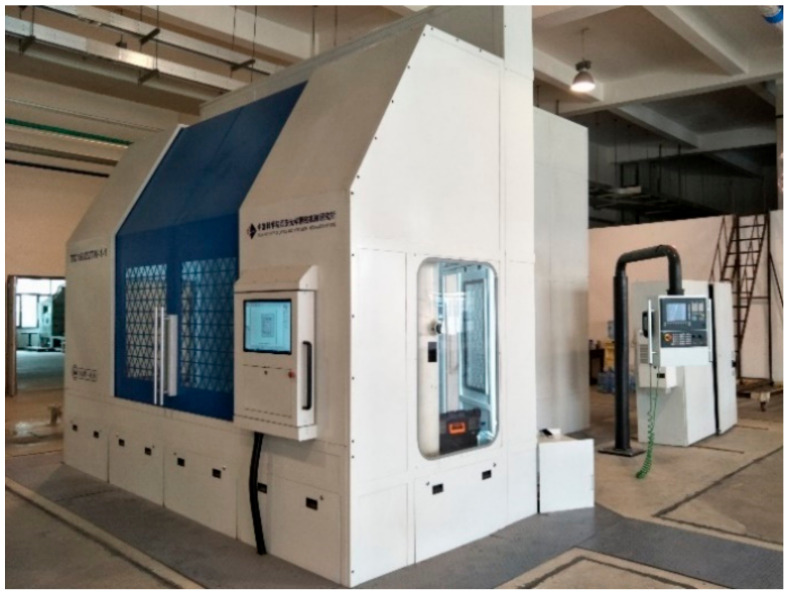
Design sketch of laser scribing equipment.

**Figure 11 micromachines-13-00323-f011:**
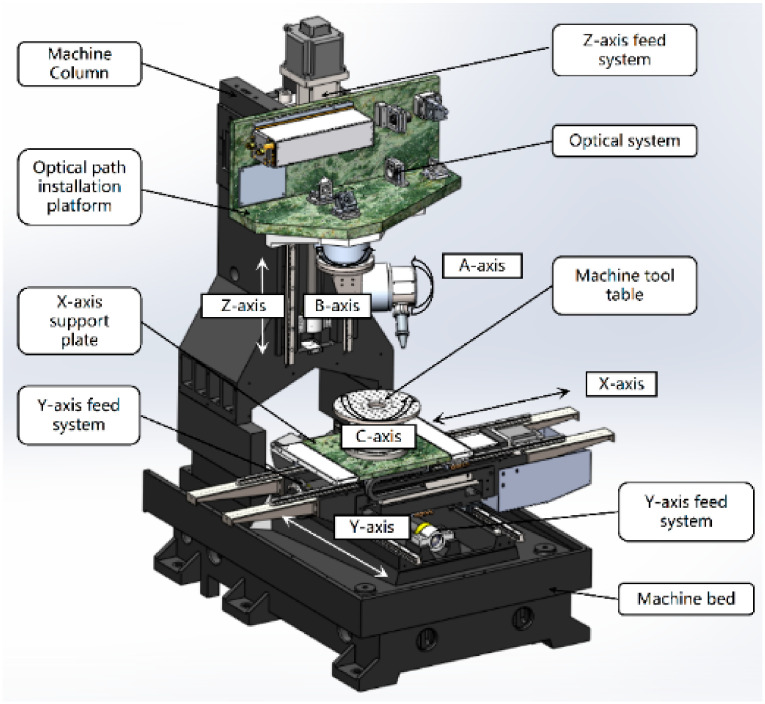
Design scheme of laser scribing machine.

**Table 1 micromachines-13-00323-t001:** Comparison of laser scribing quality under argon protection and without argon (magnified 100 times).

Frequency	500 Hz	1000 Hz	5000 Hz	10,000 Hz	15,000 Hz	20,000 Hz
Without argon protection	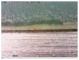	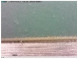	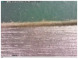	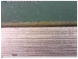	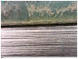	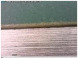
Under argon protection	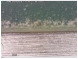	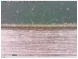	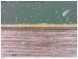	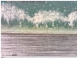	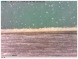	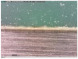

**Table 2 micromachines-13-00323-t002:** Laser scribing process parameters and scribing quality [15].

Num	Laser Type	Pulse Width (ms)	Nozzle Distance (mm)	Power (W)	Frequency (Hz)	Scribing Speed (mm/s)	Scribing Quality
1	CO_2_	0.0018	1	50	100	400	Uniform linewidth
2	CO_2_	0.0018	1	50	100	500	Uniform linewidth
3	CO_2_	0.0018	1	50	100	600	Uniform linewidth, the coating is intact

**Table 3 micromachines-13-00323-t003:** The comparative analysis of laser scribing machines.

Time	Name	Manufacturer	Application Manufacturer	Application
1980s	Multiaxis laser scribing machines	Douglas Aircraft Division of McDonnell Douglas	Douglas Aircraft Division of McDonnell Douglas	First scribing for aircraft skin
1980s	Torreslaser laser scribing machine	M. Torres, Navarra, Spanish	Airbus, Toulousem, France; Boeing, Seattle, USA	First scribing for aircraft skin
1980s	Laserdyne laser system	Prima Power Laserdyne, Minneapolis, USA	Unknown	First scribing for chemical milling parts in aerospace
1993	Automatic laser maskant scriber with linear synchronous motors (LSMs)	Boeing, Wichita, KS, USA	Boeing, Wichita, KS, USA	First scribing for aircraft skin
2010	Five-DOF gantry robotic system and flexible pogo fixture	Unknown	Unknown	First scribing for aircraft skin
2018	Laser scribing machine	Xi’an Institute of Optics and Precision Mechanics of CAS	AECC Shenyang Liming Aero-Engine Co., Ltd. Shenyang, China.	First scribing and second scribing for aerospace engine casing

**Table 4 micromachines-13-00323-t004:** The parameters of TORRESLASER laser scribing equipment.

Num	Parameter	Numerical Value
1	X/Y/Z-axis travel	15,000 mm/4000 mm/1500 mm
2	X/Y/Z-axis feed speed	10,000 mm/min
3	X/Y/Z-axis rapid traverse rate	20,000 mm/min, 20,000 mm/min, 15,000 mm/min
4	C-axis rotation range	±360°
5	C-axis rotation speed	5400°/min
6	W-axis travel	±8 mm

**Table 5 micromachines-13-00323-t005:** The parameters of laser scribing machine.

Num	Parameter	Numerical Value
1	X/Y/Z-axis travel	1000 mm/1000 mm/2000 mm
2	A/B/C-axis travel	±120°/±360°/±360°
3	X/Y/Z-axis accuracy	≤10 μm
4	A/B/C-axis accuracy	≤10″
5	C-axis load capacity	1000 kg

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
