# Peer review of "Research Status and Prospect of Laser Scribing Process and Equipment for Chemical Milling Parts in Aviation and Aerospace"

_micromachines, 2022, doi:10.3390/mi13020323_

Round 1
Reviewer 1 Report
The authors are presenting a review study on laser scribing. The paper was written using a very educational approach that I think that may help readers in understanding the process and related research.
I find some minor issues in the manuscript: space missing between numbers and units (SI), commas in p.6 - 190, use and no use of subscripts for CO2, symbols in italics, etc. Please, revise also wavelength units in p. 9.
Regarding Fig. 5, please revise the numbers. They look strange.
Regarding sections 3 and 4, it would be useful to add tables and/or images summarizing the most important results presented in these sections (materials, parameters, etc.).
Reviewer 2 Report
The paper is an experimental one, with novelty and interesting subject.
From theoretical point of view the manuscript can be improve; by considering a more realistic Lambert-Beer law. If so, the manuscript could be considered for publication.

Reviewer 3 Report
Major corrections are required regarding equations:
Eq.(1) - F0 is the absorption energy ( In Eq(3) is the fluence), equal to absorbed intensity (1-R)I0????, with "beta" called absorption coefficient (like alfa in Eq.(4)) ???? Correctly should be absorbed intensity or fluence, and beta is the absorbance
Eq.(3) - F is the energy density (laser fluence), below - I denotes the same quantity
Eqs.(4,5) work for semitransparent materials, not metals, with
" I0 ( W / cm2 ) is energy density threshold" - I could be energy density, but units should be J/cm2, not W/cm2. Here, I looks more like laser intensity/power density
Eq.(6) - f is the repetition rate
Eq.(8) - S should be not pattern spacing, but focal spot (similarly to Eq.(6))
Hence, all used quantities should be correcly called and used in Equations, while Eqs.(4,5) shuld be specified for semitransparent materials
Round 2
Reviewer 3 Report
OK, now minor corrections are required:
1) Once you use Fth in the denominator in Eq.3, it should be multipled by absorbance, since absorbed fluence F0 is used in the numerator. The same is true in Eq.5
2) In Eq.5 h is also F0-dependent quantity, not constant, and should be given as a function of F0
3) Why laser energy denisty J is used, once I0 and F0 are already introduced
